# DRAM-like Architecture with Asynchronous Refreshing for Continual Relation Extraction

## ABSTRACT

Continual Relation Extraction (CRE) has found widespread web applications (e.g., search engines) in recent times. One significant challenge in this task is the phenomenon of catastrophic forgetting, where models tend to forget earlier information. Existing approaches in this field predominantly rely on memory-based methods to alleviate catastrophic forgetting, which overlooks the inherent challenge posed by the varying memory requirements of different relations and the need for a suitable memory refreshing strategy. Drawing inspiration from the mechanisms of Dynamic Random Access Memory (DRAM), our study introduces a novel CRE architecture with an asynchronous refreshing strategy to tackle these challenges. We first design a DRAM-like architecture, comprising three key modules: perceptron, controller, and refresher. This architecture dynamically allocates memory, enabling the consolidation of well-remembered relations while allocating additional memory for revisiting poorly learned relations. Furthermore, we propose a compromising asynchronous refreshing strategy to find the pivot between over-memorization and overfitting, which focuses on the current learning task and mixed-memory data asynchronously. Additionally, we explain the existing refreshing strategies in CRE from the DRAM perspective. Our proposed method has experimented on two benchmarks and overall outperforms ConPL (the SOTA method) by an average of 1.50% on accuracy, which demonstrates the efficiency of the proposed architecture and refreshing strategy.

## KEYWORDS

Continual Relation Extraction, Dynamic Random Access Memory, Memory Allocation, Refreshing Strategy

## 1 INTRODUCTION

In pursuit of high-quality analysis of the exploding textual knowledge and construction of web applications, such as web knowledge graphs[1, 22], relation extraction attempts to automatically extract relations between two entities in a text. For example, given the text "*On January 4, 1643, Isaac Newton was born in a small village in England*" and the entity pair ("*Isaac Newton*", "*England*"), the relation extraction model should extract the relation "was born in", and this underlying capability allows it to underpin a wide range of downstream tasks [6, 31, 37].

Most of the traditional relation extraction methods focus on extracting a given set of predefined relations [10, 17, 24], and it plainly limits the usage of these methods in practical applications, where new relations keep emerging in the real world. The demands of real-world drive predecessors to pioneer practical continual learning settings [32, 33], which have been used in open learning scenarios to form the paradigm of continual relation extraction (CRE) [9, 34, 35]. CRE is considered as an adaptive algorithm for learning

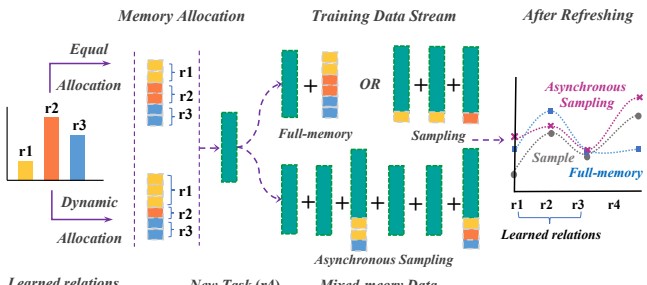

**Figure 1: Differences between traditional continual relation extraction methods and our approach. Top: traditional methods allocate equal memory samples for seen relations with different performances and use either a centralized or distributed refreshing strategy. Bottom: Our approach dynamically allocates memory for different relations according to performance and applies asynchronous refreshing to learn the current task while retaining the memory of learned knowledge. The curves on the right show the performance for all relations after learning a new relation (r4) and refreshing, and it can be observed that the asynchronous refreshing strategy is superior compared to the other two methods.**

a series of tasks including different new relations and maintaining memory of learned knowledge. Compared to conventional RE tasks, CRE tends to obtain a more stable understanding on both emerging and learned relations [4]. However, it suffers from the generic problem of catastrophic forgetting of continual learning [8, 20, 21, 28], in which knowledge learned from previous tasks is abruptly forgotten when learning from new observations. A large number of studies [4, 15, 36] have attempted to solve this problem and the main approaches can be categorized into regularization methods [38], dynamic structure methods [7], and memory-based methods [13, 40]. In the field of CRE, memory-based approaches are considered promising, which alleviate forgetting by storing memory samples of seen relations and using a refreshing strategy (also named a replay strategy). Recent studies [3, 27] have pivoted towards adapting these methods to few-shot learning scenarios.

While memory-based approaches utilize memory refresh to partially alleviate catastrophic forgetting, their memory processing predominantly relies on static storage, which applies an average allocation of memory samples. The allocation of the same sample size to all seen relations merely promotes the consolidation of well-remembered relations but hinders the revisitation of poorly remembered relations. Fig. 1 illustrates equal allocation for each relation in the upper left part, which is contrast to the inherent imbalance in learned relations, presenting a formidable challenge in achieving the allocation of memory samples. Moreover, most of

the extant methods use a monotonic refresh strategy. The approach [34] refreshes full memory alone can potentially lead to the model's detachment from acquiring knowledge, referring to the phenomenon called over-memorization. Conversely, some other methods [9] employ an excessive integration of memory refreshing with training samples which may result in over-fitting. As shown in the upper right part of Fig. 1, these monotonous refreshing strategies make it difficult for the model to maintain the memory of learned knowledge while observing new tasks.

To address the aforementioned challenges, we introduce an innovative solution involving Dynamic Random Access Memory (DRAM) in the context of a CRE scenario. DRAM is a memory hardware capable of dynamic random data access. However, its physical characteristics give rise to the issue of leakage current, which entails the gradual loss of charge over time. This phenomenon can result in the abrupt loss of data in DRAM when a certain threshold is exceeded. To mitigate this problem, DRAM employs various refreshing mechanisms (e.g. asynchronous refreshing) to replenish the charge and prevent data loss. Notably, the loss of charge in DRAM parallels the gradual forgetting of previously learned parameters in the CRE model. Both scenarios involve memory refreshing as a solution, making the incorporation of DRAM a natural solution to the challenges previously discussed.

Incorporating DRAM into CRE scenarios, we propose a DRAM-like Architecture with Asynchronous Refreshing (DAAR) to extract relations in the continual few-shot learning setting. Our method can be broken down into two key components: the DRAM-like architecture and the asynchronous refreshing strategy. (1) DRAM-like Architecture: To address the conflict between an inherent imbalance in learned relation and the unbiased allocation of an equal number of memory samples to seen relations, we devise a DRAM-like architecture consisting of three modules: the perceptron, controller, and refresher. These modules serve to quantify and transform the intrinsic imbalance in relational memory into dynamic memory sample storage, as depicted in the lower part of Fig. 1. (2) Dealing with the diversity of refreshing methods and the issues of over-memorization and over-fitting in existing approaches is another challenge. Our designed asynchronous refreshing strategy in CRE alleviates the problem of over-memorization and over-fitting by focusing on current task learning and asynchronously training mixed-memory data. Asynchronous refreshing has also been proven in experiments to be effective on several models. Meanwhile, we explain the existing refresh strategies in CRE DRAM, which are categorized into centralized, distributed refresh, and hybrid refreshing.

In summary, the contributions of this work are as follows:

- We bridge the DRAM mechanism with CRE scenario. Our innovative proposal introduces a DRAM-like architecture, which effectively addresses the challenge of relational memory imbalance and dynamically allocates memory samples.
- Different from traditional memory refreshing, we devise an asynchronous refresh strategy for guiding the refreshing of dynamic memory samples and further provide a theory that explains existing refresh strategies.
- Extensive experiments are carried out on two benchmarks, *i.e.* FewRel [11] and TACRED [39], where our method achieves promising relation extraction results for each task.

## 2 RELATED WORK

In this section, we summarize the literature reviews in three main areas that are related to the study:

**Continual Relation Extraction.** Continual relation extraction is proposed to address the problem of growing relations in the real world. Past research in this domain can be broadly delineated into three primary strategies: (1) Regularization methods ([15, 18, 29, 38]), which constrain the updates to the neural weights of preceding tasks. (2) Dynamic structural methods, introducing structural modifications like module additions to learn new tasks without compromising previously acquired knowledge. Notable examples include [7, 30, 36]. (3) Memory-based methods have proven promising in the field of natural language processing [2, 4, 13, 40], which prevents catastrophic forgetting by selectively archiving and refreshing samples from earlier tasks. For instance, RP-CRE [4] refines subsequent sample embeddings by the prototype of all observed relations. EDRA [27] incorporates embedding space regularization and data augmentation to handle the incompatibility. Notwithstanding their merits, a common limitation of these methods is their reliance on static memory, leading to an arguably inequitable consolidation of memories. In contrast, our proposal towards a dynamic memory consolidation approach.

**Refreshing Strategy.** We offer a novel framework for classifying memory-based models based on refreshing strategies, encapsulating existing models into three overarching paradigms: (1) Centralized refreshing was used often in slightly earlier work [14] to create an isolated memory batch designated for centralized replay. More contemporary studies, such as those by Wang et al. [34], continue to harness this approach. (2) Distributed refreshing integrates memory across the extents of individual training data sets. Earlier GEM [19] dispersed the previous task to a new task by constraining the gradient to learn a subset of correlations common to a set of distributions. After this, AGEM [2] optimizes it. In recent years, Qin et.al. [27] have similarly used distributed refreshing to replay episode memories. (3) Hybrid refreshing is generally a mix of centralized and distributed refreshing such as asynchronous refresh, etc. Chen et al. [3] update the prototype parameters by the centralized refreshing strategy and adept distributed refreshing strategy for encoder parameters. In contrast to the over-memorization problem of centralized refreshing and the overfitting of distributed refreshing, Experimental results substantiate the capability of asynchronous refreshing strategy to bridge the aforementioned refreshing paradigms effectively.

**Analogical Modeling.** Analogical modeling methods are a slightly broader topic and are widely used in various disciplines. Here, we describe only some analogical modeling methods related to computers and artificial intelligence. Analogical modeling approaches draw on the laws and mechanisms of natural ecosystems [23, 25], abstracting the problem to the interactions and effects of the various elements of the ecosystem. Other approaches, grounded in tangible physical and mathematical processes [12, 16], recast problems to align with the underlying laws and models of these domains. Cross-disciplinary analogical modeling merges insights and techniques from varied fields. For example, [9] introduced episodic memory activation and reconsolidation for CRE tasks inspired by human long-term memory formation. In this paper, the structure

built by analogy to DRAM can effectively perceive the forgetting and thus dynamically allocate memory.

## 3 METHODS

### 3.1 Problem Definition and Background

This section includes the generic problem definition and external information to provide background knowledge about Dynamic Random Access Memory (DRAM).

**Problem definition.** In the context of CRE, we consider a sequence of $n$ tasks, denoted as $\{\mathcal{T}_1, \mathcal{T}_2, ..., \mathcal{T}_n\}$. Each task $\mathcal{T}^k$ ($k \in [1, 2, ..., n]$) is a few-shot supervised learning task comprising training, validation, and testing datasets, symbolized as $C_{\text{train}}^{(k)}, C_{\text{valid}}^{(k)}, C_{\text{test}}^{(k)}$, respectively. Each dataset contains relation set $R^{(k)}$ delineating the relations within and labeled instances $\{s_i, y_i\}$, where $y_i$ is a relation label corresponding to the sentence $s_i$. The aim is to architect a versatile classification model, $f$, capable of observing task $\mathcal{T}^{(k)}$ in step $k$ and adeptly handling the antecedent $k - 1$ tasks. Performance of this model $f$ is evaluated on the test sets $\{C_{\text{test}}^{(1)}, ..., C_{\text{test}}^{(k-1)}\}$, aiming to capture the effect of model memorization on $k - 1$ tasks.

Different from previous well-performing memory-based work [3, 4, 27], we introduce a dynamic episodic memory framework, denoted as $\mathcal{M} = \{\mathcal{M}^1, \mathcal{M}^2, ...\}$, where each $\mathcal{M}^k$ is constructed from the corresponding task $\mathcal{T}^k$. Notably, the number of samples stored for each relation within $\mathcal{M}^k$ is variable. This design is inspired by the human cognitive process, where attention is selectively allocated to different memories, thereby introducing a bias towards relations that have historically poor-trained.

**DRAM.** As shown in Fig. 2, memory cells in a DRAM array are arranged in a rectangle, and each memory cell stores 1 bit of data. DRAM cell consists of a capacitor and an access transistor. Due to the physical properties of the capacitor, leakage current is generated and causes the charge on the capacitor to be lost over time. When this charge dwindles below a critical threshold, the DRAM's ability to discern is compromised, ultimately resulting in data corruption. The detection of leakage current of contemporary DRAM cells involves a synergistic interplay between lasers and detectors. Upon surpassing the established threshold, the leakage current is transmitted to a controller, prompting a refresh command. During the DRAM refresh process, the original data is first read, the capacitor level is compared with the reference level to determine the 1/0 value of the data, and then the original data is written back, which is like a memory replay operation.

The working mode of DRAM is similar to CRE tasks in many aspects, such as leakage current and catastrophic forgetting, same refreshing strategies, etc. Drawing parallels from these observations, we devised a DRAM-like structure using analogy modeling, as elucidated in Fig. 2.

### 3.2 DRAM-like Architecture

Drawing inspiration from DRAM, we propose a novel architecture endowed with the capability of dynamically replaying memory while maintaining memorization of previous learned knowledge in the domain of continual learning. This advancement augments memory utility, fortifying the learning process as new tasks emerge.

**Based Module.** (1)*Encoder.* we use BERT [5] as the base model which feeds a sentence $s$ with a head entity $e_h$ and a tail entity $e_t$. To enhance the representation of input sentences, we adept the specific input from [3], which is described as $s_{\text{input}} = \{ [\text{CLS}], e_h, [\text{MASK}], e_t, [\text{SEP}], s, [\text{SEP}] \}$. We can obtain the contextualized representation of input sentences. The [MASK] token can be considered as the relational representation. (2)*Prototype classifier with memory.* prototype is initialized by the aggregation of all current task samples and updated in the subsequent training process. Specifically, the current task $C_{\text{train}}^{(k)} = \{C_t^{r_1}, C_t^{r_2}, ..., C_t^{r_N}\}$ which describe the $k^{th}$ task train set with $N$ relations set $R_{\text{train}}^k = \{r_1, r_2, ..., r_N\}$, is aggregated to calculate the prototypical representation $\mathcal{P}_j$ of each class.

$$\mathcal{P}_i = \frac{1}{|C_t^i|} \sum_{s_i \in C_t^i} \text{Encoder}(s_i) \tag{1}$$

where $C_t^{r_i} \in C_{\text{train}}^{(k)}$ is train data of the $i^{th}$ relation in the $k^{th}$ task train set and $s_i$ is the instance in $C_t^{r_i}$. We can obtain the current prototype representation $\mathcal{I}^k = \{\mathcal{P}_1, \mathcal{P}_2, ...\}$ corresponding the current task $C_{\text{train}}^k$. Upon the arrival of a new task, prototypes $\mathcal{I}^k$ are initialized; subsequently, these prototypes $\mathcal{I}^k$ are updated only in the training process with new task data and through memory replay. Different from the approach delineated in [3], our approach does not employ prototype memory, thus providing more room to increase the number of samples memorized for each relation.

**Perceptron.** The perceptron module plays a pivotal role in our setup, adept at evaluating the impact of each observed relation, and further generating a perceptive message via linear transformation. As we anticipate the arrival of a task $\mathcal{T}^k$, the perceptron takes the initiative to procure a test dataset encompassing all previously observed relations, denoted as $C_{\text{past}}^{k-1} = \{C_p^{r_1}, C_p^{r_2}, ..., C_p^{r_N}\}$. Within this context, $N$ signifies the size of the historical relation set $R_{\text{past}}^{k-1} = \{r_1, r_2, ..., r_N\}$. For any relation $r \in R_{\text{past}}^{k-1}$, we obtain the prototype $\mathcal{P}_r$ as the relation representation, along with the sample set $C_p^r$. Assuming the encoder $f(\cdot)$ with parameter $\theta$, the perceptual score $F_r$ for the selected relation $r$ can be calculated using the following equation:

$$F_r = \frac{1}{|C_p^r|} \sum_{(s_i) \in C_p^r} \text{Sim}\left(\mathcal{P}_r, f(s_i)_\theta\right) \tag{2}$$

where $\text{Sim}(\mathcal{P}_r, f(s_i)_\theta)$ is the vector dot product for relation representation and instance feature. Then perform this operation on the set of seen relations $R_{\text{past}}^{k-1}$ and concatenating all the results gives $\mathbf{F} = [F_1, F_2, ..., F_N]$, characterizing how well the set of historical relations performs on the model. In order to scale up the numbers for those relations that perform poorly, we employ the softmax function, subsequently transforming $\mathbf{F}$ with a uniform vector, $\mathbf{E}$. The result perceptive message $Mes$ will be sent to the controller for assigning memory.

$$Mes = \text{Softmax}\left(\mathbf{E} - \mathbf{F}\right) \tag{3}$$

**Controller.** Before the perceptive message $Mes$ is sent, the controller engages in crucial preparatory steps comprising memory expansion and selecting informative samples. In the process of memory expansion, given the conservation of $E_N$ samples for each

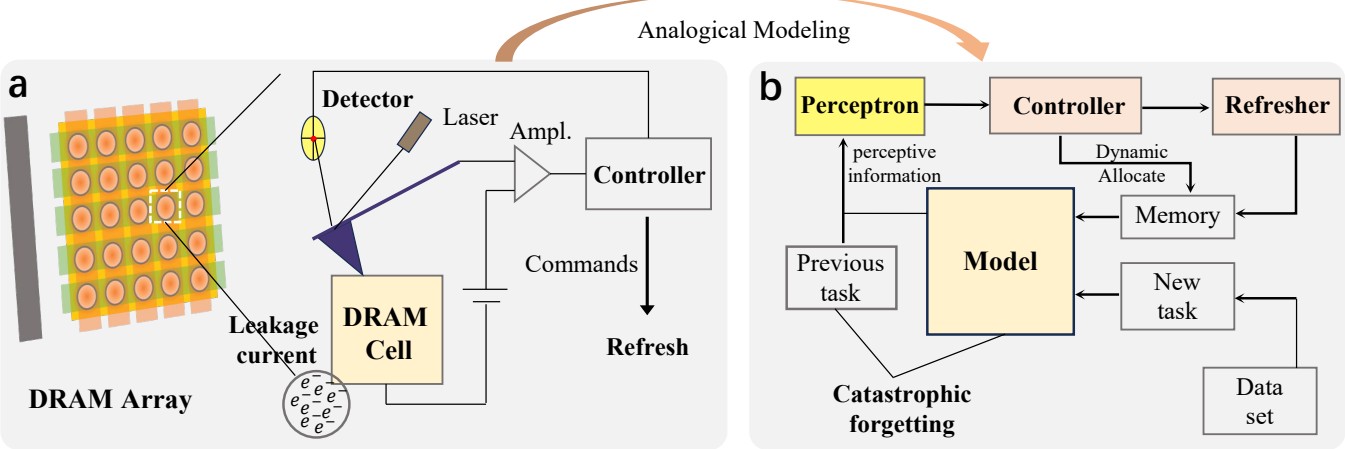

**Figure 2: Continual relation extraction model framework inspired by the DRAM structure. Left: Architecture of DRAM cell leakage current and refresh circuits in DRAM arrays. Right: The model framework obtained through analogy modeling with DRAM, in which the catastrophic forgetting is similar to the leakage current. The main workflow is: after sensing the model's learning performance on previous tasks using perceptrons, the controller receives the information and dynamically allocates memories, issuing commands that cause the refresher to refresh the memories. The model learns the new task and reviews the learned knowledge using mixed-memory data.**

relation, the memory size $|\mathcal{M}|$ will be expanded to:

$$|\mathcal{M}| = |\mathcal{M}| + E_N \cdot |R_{\text{train}}^k| \quad (4)$$

$R_{\text{train}}^k$ represents the relation set of current training task. All samples relating to a given relation $r$ are systematically ranked in descending order based on their proximity to the prototype representation $\mathcal{P}r$. This furnishes a set of informatively ranked samples: $C\text{sorted}^r = \hat{C^{r_1}}, \hat{C^{r_2}}, ..., \hat{C^{r_N}}$. Upon receipt of $Mes$, the controller promptly derives the requisite number of samples for memory allocation for each relation:

$$\mathbf{Q} = (E_N \cdot |R_{\text{train}}^k|) \odot Mes \quad (5)$$

where $\mathbf{Q} = \{q_1, q_2, ...q_N\}$ is the storage vector with $q_i$ symbolizing the sample count allocated to the $i^{th}$ relation. $\odot$ represents the dot product of vectors. Subsequently, the top $q_i$ samples from each set, pertaining to relation $r_i$, are denoted as $\hat{C^{r_i}}q_i$. The memory storage relation for the current task is thus formulated as:

$$\mathcal{M}_A = \left\{ \hat{C}_{q_i}^{r_i}, 1 \le i \le N, \forall r \in R_{\text{train}}^k \right\} \quad (6)$$

$\mathcal{M}_A$ represents the memory assignment in the current task $\mathcal{T}^k$, then the new memory is updated to $\mathcal{M} = \mathcal{M} + \mathcal{M}_A$. Afterward, the controller has the additional task of setting the appropriate size of the memory cells and the frequency of replays for subsequent memory refreshing. The controller sends the refreshing strategies stored internally to the refresher and waits for the next round of tasks to be processed.

**Refresher.** When the $k^{th}$ task arrives, the refresher is immediately activated, processing parameters relayed from the controller, and then adopt the refreshing method according to the delineated strategy. After that, the refresher starts the update module of the prototype and performs the memory refresh. We have built in three strategies of centralized, distributed, and asynchronous refreshing

in the refresher, but in order to ensure fairness, the total amount of memory used by these three strategies is the same.

## 3.3 Refreshing strategy

This section introduces three refreshing strategies, which attempt to expound existing memory-based methods and unify them within this theoretical system. When the $k^{th}$ task $\mathcal{T}^k$ arrives at refresher, the total memory $\mathcal{M}$ of the previous k-1 tasks has already been acquired through the dynamic storage mechanism. The refresher further extracts the training dataset, $C_{\text{train}}^k$, associated with $\mathcal{T}^k$. Subsequent action involves model training with memory refreshing in $\mathcal{E}$ epochs. The mixed-memory data for this purpose is represented as $\mathcal{D}_{\text{mix}}$. We define the three refresh strategies as follows:

**Centralized refresh** distinctly earmarks a specified duration within the training phase exclusively for memory refreshing. When centralized refreshing starts, the refresher acquires the predefined number of memory refreshing epochs $h$, and then divides the total amount of memory $\mathcal{M}$ to obtain the centralized refreshing memory size $|\mathcal{M}_c| = |\mathcal{M}|/h$. Over time, if current epoch $j$ is smaller than the total number of epochs $\mathcal{E}$, the mixed-memory data $\mathcal{D}_{\text{mix}}$, align with $C_{\text{train}}^k$. Conversely, when the epoch $j$ surpasses $\mathcal{E}$, then $\mathcal{D}_{\text{mix}} = \mathcal{M}_c^i$ where $\mathcal{M}_c^i$ represents the $i^{th}$ block of $\mathcal{M}$ with size $|\mathcal{M}_c|$ memory.

**Distributed refresh** in DRAM is the process of putting the replenishment of charge into the cycle after each read or write. This is similar to how the distributed refreshing strategy mixes memories into each training data before training. When the refresher captures distributed refreshing strategy, without setting additional parameters but directly slices the total amount of memory $\mathcal{M}$ into memory cells $\mathcal{M}_d$ according to the total number of epochs $\mathcal{E}$. The mixed-memory data for epoch $j$ is denoted as $\mathcal{D}_{\text{mix}}^j = C_{\text{train}}^k + \mathcal{M}_d^j$, subsequently serving as the direct input data for training the model.

The adoption of **asynchronous refresh** strategies has become commonplace in the contemporary DRAM space. This strategy strikes a balance between excessive refreshing and dedicating an extensive block of time for refreshing. Our proposed asynchronous refreshing requires the refresher to set an additional parameter $\mathbb{I}$, representing the interval for memory refreshing operations. Then the refresher will give the mixed-memory data $\mathcal{D}^l_{\text{mix}} = C^k_{\text{train}} + \mathcal{M}^l_a$ for $l \in [\mathbb{I}, 2\mathbb{I}, ...]$. $\mathcal{M}^l_a$ represents the $l^{th}$ block of $\mathcal{M}$ with size $\frac{|\mathcal{M}|}{\lfloor \mathcal{E}/\mathbb{I} \rfloor}$ memory. In epochs other than these specific intervals, the mixed-memory dataset defaults to $\mathcal{D}_{\text{mix}} = C^k_{\text{train}}$. To the best of our knowledge, asynchronous refresh is the first proposed in terms of memory refreshing strategies for continual learning.

### 3.4 Learning Procedure

Before initiating the training of the $k^{th}$ task, the perceptron module evaluates the performance of previously seen relations on the current model. Subsequently, it linearly transforms this evaluation into a perceptive message that is communicated to the controller. Once the controller completes preparatory tasks such as memory expansion, it dynamically allocates memory based on the perceptive message and generates a new memory set denoted as $\mathcal{M}$. When the $k^{th}$ task arrives and the prototype representation $\mathcal{I}^k = \mathcal{P}1, \mathcal{P}2, ...$ is initialized, the refresher module combines memory cells to create mixed-memory data $C\text{mix}^k = C\text{train}^k + \mathcal{M}$ according to the selected refreshing strategy. The model is then trained using a cross-entropy loss function to learn the consistency between the old and new distributions. For any $s \in C^k_{\text{mix}}$, $\mathcal{P}$ represents the relational representation corresponding to $s$ in $\mathcal{I}^k$. Assuming $P$ represents the true distribution of $C^k_{\text{mix}}$, the loss function is defined as follows:

$$\mathcal{L}^C(\theta) = \mathbb{E}_{s \sim P} \left[ -\log Q \left( \text{Sim}(f_\theta(s), \mathcal{P}) \right) \right] \quad (7)$$

where $Q(\text{Sim}(f_\theta(s), \mathcal{P}))$ represents the relational distribution of model output. To further distinguish similar prototype representations, we also designed the contrastive loss function. For any $\mathcal{P} \in \mathcal{I}^k$, assuming that $\Omega$ is the prototype representation containing $\mathcal{P}$ and those similar to $\mathcal{P}$, and $\Gamma$ is the expected one-hot distribution generated from $\Omega$, the contrastive loss function can be defined as:

$$\mathcal{L}^{\mathcal{A}}(\theta) = \mathbb{E}_{\mathcal{P} \sim \Gamma} \left[ -\log \Omega(\mathcal{P}) \right] \quad (8)$$

Assuming that $\lambda_1$ and $\lambda_2$ are the weighting coefficients of the above two loss functions, we can obtain the final loss function as:

$$\mathcal{L}(\theta) = \lambda_1 \cdot \mathcal{L}^C(\theta) + \lambda_2 \cdot \mathcal{L}^{\mathcal{A}}(\theta) \quad (9)$$

In the inference phase, we obtain the encoded representation of the sentence and compute the distance matrix with the prototype representation. From the distance matrix, we acquire the type of relation to the model output, which is compared with the label, and used to compute the final whole accuracy.

## 4 EXPERIMENTS

### 4.1 Benchmark and Evaluation Metric

In this section, we present the datasets used in our experiments and the evaluation metrics.

**Benchmark.** Our experiments are conducted on two well-established benchmarks, in accordance with prior research work [3]: (1) **FewRel** [11] is a large-scale dataset that contains 100 relations, each of which has 700 instances. Following NK-CRE [3], we use the publicly accessible 80 relations in the training and validation sets into 8 tasks containing 10 relations (10-way). In order to align the SOTA method [3], we carried out the experiment 10-way-5-shot on the FewRel Benchmark. (2) In addition to demonstrating the generalizability of our paradigm, we also conduct experiments on **TACRED**, which is a RE dataset proposed by [39]. Different from FewRel, it contains 42 relations and over 100,000 instances. In NK-CRE, it remains 41 relation classes and 68,438 instances after filtering out the relation "n/a". In this paper, we also conduct the experiment 5-way-5-shot on the TACRED to illustrate the applicability of our methodology.

**Evaluation metric.** At time step $k$, we first acquire the test sets $\hat{C}^k_{\text{test}} = \bigcup_{i=1}^k C^i_{\text{test}}$ of all seen tasks $\{\mathcal{T}^i\}^k_{i=1}$. Then evaluate the model performance on $\hat{C}^k_{\text{test}}$ with the **whole accuracy**. It can be defined as:

$$ACC_{whole} = acc_{f, \hat{C}^k_{\text{test}}} \quad (10)$$

Owing to the whole test set of all tasks used to calculate the accuracy, it actually reflects the model's ability to alleviate catastrophic forgetting while effectively assimilating novel knowledge.

### 4.2 Baselines

DRAM-like architecture with an asynchronous refreshing strategy is to extract relation with continual learning. Given that recent models have not employed the centralized refresh mechanism, our experiments also revealed its inadequacy in this context. Consequently, we opted not to include the centralized refresh baseline for comparison. The compared baselines are set as follows:

**EMAR**: Episodic memory activation and reconsolidation (EMAR) is a pioneering method [9] to alleviate the problem of catastrophic forgetting in continual relation learning. EMAR uses relation prototypes for memory reconsolidation exercise to maintain a stable understanding of old relations while learning new ones.

**ERDA**: ERDA [27] is an innovative method that defines the formulation of the challenging problem of continual few-shot relation learning (CFRL). They propose a novel method that incorporates embedding space regularization and data augmentation to handle the incompatibility between feature distributions of new and previous tasks.

**ConPL**: A current SOTA method [3] for N-way-K-shot Continual Relation Extraction (NK-CRE) task. ConPL consists of three modules: prototype-based classification, a memory-enhanced module for vital sample selection, and a consistent learning module to alleviate catastrophic forgetting.

The method proposed in this paper can regulate memory storage by changing the $E_N$ parameter in Eq. 4. In order to investigate the effect of memory size on forgetting, we conducted memory expansion experiments on both ERDA and ConPL. For EMAR, we cannot extend the investigation due to data limitations. The performance of EMAR in the table is derived from [3].

### 4.3 Implementation Details.

Our experimental setup was conducted on a single NVIDIA 3090 GPU utilizing the PyTorch framework [26]. Our DRAM-like architecture leverages $BERT_{BASE}$ [5] as the backbone encoder, with the base prompt template being e1 [MASK] e2. Due to hardware constraints, we employed a batch size of 4. Gradient updates were facilitated using the Adam optimizer, initialized with a learning rate of $2e^{-5}$.

In addition, the number of samples $E_N$ that are kept in memory of Eq. 4 is set to 2. The loss weights, denoted as $\lambda_1$ and $\lambda_2$, are both set to 1.0. As part of our refreshing strategy, we conduct a total of $\mathcal{E} = 6$ epochs, with an asynchronous refreshing interval denoted as $\mathbb{I} = 2$. Notably, we re-performed the experiments for ERDA and ConPL with memory samples of 1 and 2 for each relation under the equal memory samples for eight tasks for a fair comparison.

### 4.4 Main Results

The whole accuracy (%) across different methods, along with their extended experiments on the two benchmarks is presented in Tab. 1. We can observe that:

(1)With equal memory samples, DAAR outperforms on both benchmarks. DAAR approaches the first task as a standard relation extraction task; hence, the number of retained relation samples does not impinge on its performance of the first task. For the eighth task, our method achieves 86.07% and 87.71% when storing 1 and 2 samples on FewRel, and 75.59% and 77.89% when storing 1 and 2 samples on TACRED. It can be observed that DAAR still maintains higher performance than ConPL, especially when the storage relation sample is 2. Although the performance of DAAR on the first task of TACRED 95.17% is slightly lower than the performance of 97.89% achieved by ConPL when the storage relation sample is 2. DAAR's dynamic memory and asynchronous refresh quickly play a role in making forgetting speed significantly slowed down, and the second task can still lead the performance by 2.28% when the performance of the first task lags 2.72%. Besides, our method continues to maintain superior and smooth performance in subsequent tasks. As shown in Figure 3, DAAR maintains the advantage of slow forgetting compared to other methods on most tasks.

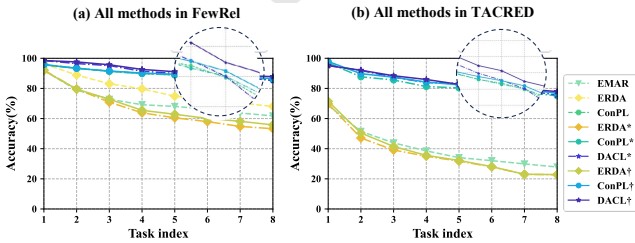

**Figure 3: Whole accuracy (%) of the different methods after training on a series tasks of FewRel benchmark and TACRED benchmark.**

(2) Increasing the number of memory samples corresponding to each relation helps reduce forgetting of the model. For ERDA, the performance improvement is not obvious when increasing the number of relational memory samples on the first two tasks. Nevertheless, when the model learns more tasks, larger memory samples significantly improve the performance of subsequent tasks. ConPL has also been improved by increasing the number of memory samples. On FewRel, from a slight improvement in the first task to a 1.25% performance improvement in the last task. However, the impact is more obvious on DAAR. On the eighth task of TACRED, the performance increased by 2.3%. On FewRel, there was also a 1.74% performance improvement. For DAAR, the reason for this result is more relational memory samples increase the allocation space for dynamically allocated memory operations performed by the perceptron and controller working together, *i.e.* for those relations with poor performance, more memory samples can be allocated for consolidation. In Figure 3, it's evident that the method with more memory samples markedly outperforms the one with fewer samples.

### 4.5 Ablation Study

We conducted two ablation experiments on FewRel, including mixed multi-style module experiments and cross-model refreshing experiments to verify the effectiveness of each module in our proposed DRAM-like architecture and the generalizability of the asynchronous refresh strategy.

**Mixed multi-style module experiments.** In order to investigate the effectiveness of each module in the DRAM-like architecture, we add the perceptron, controller, and refresher to the architecture one by one, while loading each of the three refreshing strategies with the refresher to validate their effects, as shown in Tab. 2. In order to remove certain modules while the remaining modules still work, we use some methods from ConPL[3] on each auxiliary task to complement the removed modules. Pro⁻ uses the basic prototype learning module with memory replay from ConPL, while Pro(C) represents our prototype learning module with a centralized refreshing strategy and D, A denotes distributed refreshing and asynchronous refreshing, respectively. The difference between Pro⁻+P and Pro⁻+PC is that the former perceives the information and then proceeds to distribute the relational memory samples equally, whereas the latter utilizes the perceived information to dynamically distribute the memory samples. From the first three rows of Tab. 2, it is evident that dynamic memory allocation, facilitated by the cooperative functioning of the perceptron and the controller, yields the highest overall accuracy. Examining the last three rows of 2, it can be argued that the asynchronous refresh strategy further enhances performance within the same DRAM-like architecture. A comparative analysis between the first three rows and the last three reveals that the DRAM-like architecture holds a distinct advantage over the decentralized modules, a superiority further elucidated in Fig. 4.

**Cross-model refreshing experiments.** To verify the generalizability of the asynchronous refreshing strategy, we extend 2 relational memory samples for ConPL and EDRA and asynchronous refreshing experiments on FewRel and TACRED. The symbols in Tab. 3 represent the same meaning as in Tab. 1, with "asyn" representing the addition of an asynchronous refresh mechanism. In the FewRel benchmark, when the seed is set to 100, the first three rows of the table are the expansion experiments corresponding

**Table 1: Whole accuracy (%) of the different methods after training on a series tasks of 10-way-5-shot of FewRel benchmark and 5-way-5-shot of TACRED benchmark. The unmarked methods are directly from [3] and we reproduce results in publicly available codebases of ConPL and ERDA. * represents experiments that initially store one memory sample for each seen relation and † represents initially storing two memory samples for each seen relation. The best values on each task under the same setting are denoted in bold.**

| Method | Task Index | | | | | | | |
|---|---|---|---|---|---|---|---|---|
| | T1 | T2 | T3 | T4 | T5 | T6 | T7 | T8 |
| 10-way-5shot of FewRel | | | | | | | | |
| EMAR[9] | 92.03 | 78.87 | 72.81 | 69.19 | 68.05 | 66.23 | 63.68 | 61.77 |
| ERDA[27] | 96.38 | 88.91 | 83.10 | 79.73 | 74.83 | 72.84 | 70.28 | 68.07 |
| ConPL[3] | 95.72 | 93.53 | 91.31 | 89.95 | 88.93 | 88.39 | 87.43 | 85.77 |
| ERDA*[27] | 92.17 | 79.59 | 70.85 | 63.82 | 60.50 | 57.97 | 54.77 | 53.26 |
| ConPL*[3] | 95.65 | 93.45 | 91.36 | 89.83 | 89.00 | 88.19 | **87.52** | 85.21 |
| **DAAR*(Ours)** | **98.50** | **96.50** | **95.17** | **91.40** | **89.92** | **88.85** | 87.51 | **86.07** |
| ERDA†[27] | 91.82 | 79.52 | 72.84 | 65.48 | 62.85 | 60.08 | 58.18 | 55.79 |
| ConPL†[3] | 95.87 | 93.23 | 91.58 | 90.18 | 89.40 | 88.76 | 87.96 | 86.46 |
| **DAAR†(Ours)** | **98.50** | **97.60** | **95.70** | **92.60** | **91.10** | **90.32** | **88.66** | **87.81** |
| 5-way-5-shot of TACRED | | | | | | | | |
| EMAR[9] | 68.71 | 51.53 | 43.86 | 38.54 | 34.08 | 32.06 | 29.90 | 27.87 |
| ERDA*[27] | 69.79 | 47.11 | 39.13 | 35.01 | 31.71 | 27.94 | 22.97 | 22.77 |
| ConPL*[3] | **97.03** | 87.70 | 85.60 | 81.25 | 80.32 | 78.70 | 77.32 | 75.14 |
| **DAAR*(Ours)** | 95.17 | **91.93** | **87.48** | **83.91** | **83.34** | **81.76** | **78.78** | **75.59** |
| ERDA†[27] | 71.46 | 50.41 | 41.49 | 35.58 | 32.19 | 28.11 | 23.02 | 22.82 |
| ConPL†[3] | **97.89** | 89.79 | 87.43 | 84.20 | 82.39 | **79.96** | 79.12 | 76.93 |
| **DAAR†(Ours)** | 95.17 | **92.07** | **88.41** | **85.92** | **82.68** | 79.49 | **79.73** | **77.89** |

**Table 2: Ablation experiments on the FewRel benchmark are used to validate the effectiveness of each module. $Pro^-$ represents basic prototype learning under the hybrid refresh method used in [3]. P, C and R stand for perceptron, controller and refresher respectively. C, D, A are abbreviations for centralized, distributed and asynchronous refreshing.**

| Method | T1 | T2 | T3 | T4 | T5 | T6 | T7 | T8 |
|---|---|---|---|---|---|---|---|---|
| $Pro^-$ | 95.38 | 92.82 | 90.51 | 88.42 | 87.22 | 86.68 | 85.56 | 83.80 |
| $Pro^-$+P | 95.87 | 93.23 | 91.58 | 90.18 | 88.94 | 87.49 | 86.88 | 85.05 |
| $Pro^-$+PC | 96.63 | 94.27 | 91.98 | 90.20 | 89.39 | 88.79 | 87.98 | 86.49 |
| Pro(C)+PCR | 98.50 | 97.15 | 95.50 | 92.00 | 91.04 | 90.12 | 88.59 | 87.05 |
| Pro(D)+PCR | 98.50 | 97.35 | 95.43 | 92.20 | 90.66 | 90.03 | **88.90** | 87.81 |
| **Pro(A)+PCR** | **98.50** | **97.60** | **95.70** | **92.60** | **91.10** | **90.32** | 88.66 | **87.98** |

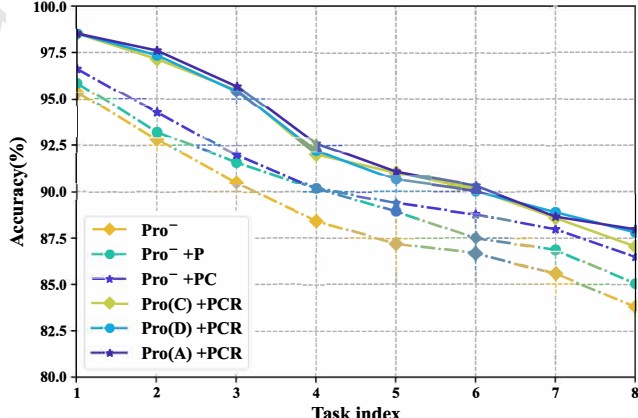

**Figure 4: The analysis result of the importance of the modules in DRAM-like structure with asynchronous refreshing strategy.**

to EDRA, and the last three rows are the expansion experiments corresponding to ConPL, we can observe that applying the asynchronous refresh strategy on the ConPL and ERDA models still allows both models to obtain some performance improvement, and also aids in verifying that multiple samples are useful for modeling to alleviate forgetting.

## 4.6 Forgetting Analysis

To further analyze the degree of forgetting of our model in continual relation extraction, we introduce an enhanced measure of prototype forgetting. This measure quantifies the forgetting of the prototype

for the $i^{th}$ task after training the $j^{th}$ task the as follows:

$$F_{i,j} = \frac{1}{|R^i| \cdot j} \sum_{k=i}^{j} \sum_{r}^{R^i} \max(0, a_{k,r} - a_{j,r}) \tag{11}$$

**Table 3: Ablation experiments on the FewRel benchmark and the TACRED benchmark(in appendix A.1) are used to validate the effectiveness of increasing the memory sample size and the asynchronous refreshing mechanism, The individual symbols represent the same meaning as in Tab. 1 and asyn denotes the asynchronous refreshing added to the method.**

| Method | Task Index | | | | | | | |
|---|---|---|---|---|---|---|---|---|
| | T1 | T2 | T3 | T4 | T5 | T6 | T7 | T8 |
| 10-way-5-shot of FewRel | | | | | | | | |
| ERDA[27] | 92.80 | 76.90 | 67.83 | 62.70 | 58.70 | 55.78 | 51.11 | 51.39 |
| ERDA$^\dagger$[27] | 92.20 | 76.50 | **71.00** | 61.30 | 63.30 | 57.00 | 53.99 | 54.13 |
| **EDRA$^\dagger$[27] + asyn** | **93.80** | **78.55** | 69.03 | **66.40** | **65.42** | **61.55** | **57.36** | **56.55** |
| ConPL[3] | 94.30 | 93.60 | 92.03 | 88.63 | 88.32 | 86.70 | 86.33 | 84.93 |
| ConPL$^\dagger$[3] | 95.90 | 93.70 | 92.20 | 88.70 | 88.34 | 87.33 | 87.60 | 86.74 |
| **ConPL$^\dagger$[3] + asyn** | **96.40** | **94.85** | **92.90** | **90.05** | **89.26** | **88.37** | **88.75** | **87.23** |

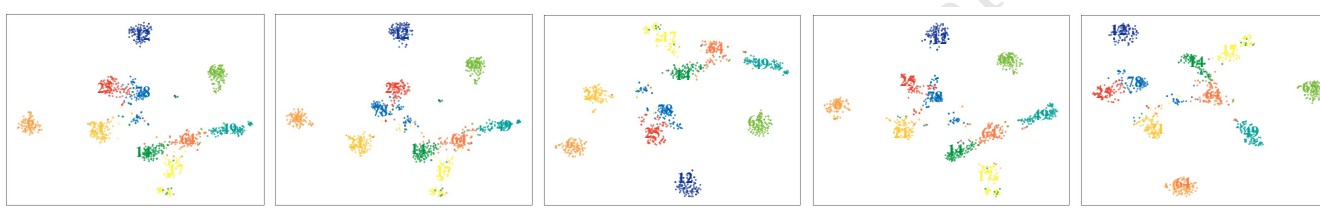

(a) t-SNE visualization(task-1)   (b) t-SNE visualization(task-2)   (c) t-SNE visualization(task-3)   (d) t-SNE visualization(task-4)   (e) t-SNE visualization(task-5)

**Figure 5: t-SNE visualization of first task features learned by DAAR at training Task-1 to Task-5 on FewRel. The first task contains a total of ten relations, and we characterize the different classes of relations with different colors.**

**Table 4: ConPL and DAAR methods of forgetting Task-1 forgetting(%) after learning Task-1 to Task-5. † represents initially storing two memory samples for each seen relation.**

| Method | Task index | | | | |
|---|---|---|---|---|---|
| | T1 | T2 | T3 | T4 | T5 |
| ConPL$^\dagger$[3] | 0.00 | 3.25 | 3.63 | 6.20 | 6.50 |
| DAAR$^\dagger$ | 0.00 | **0.95** | **3.57** | **3.18** | **5.22** |

where $a_{k,r}$ represents the accuracy on relation $r$ for the set of relations $R^i$ belonging to task $i$ after the $k^{th}$ task training. The $\max(0, a_{k,r} - a_{j,r})$ is expressed in the relation $r$, the degree of forgetting at training prior $k^{th}$ task versus subsequent $j^{th}$ task. We provide the degree of forgetting for the first task when trained sequentially on the subsequent four tasks with ConPL and DAAR, as shown in Tab. 4. We can observe that DAAR forgets a little less compared to ConPL for the same measure of forgetting level, which explains the higher accuracy of DAAR after the first task.

To better observe how DAAR learns the features of the first task when subsequent tasks are learned continuously, we used t-SNE to visualize the ten relation categories of the first task and plotted the change in features of Task-1 when learning Task-1 to Task-5, as depicted in Fig. 5. A slightly larger change that appears in Fig. 5(c) is the feature of Task-1 when the DAAR has finished learning Task-3, although only by flipping the top and bottom features. After this,

the feature of Task-1 is corrected again after learning Task-4. This phenomenon is consistent with Tab. 4, which shows an increase followed by a decrease in the DAAR's forgetting of Task-1 after learning Task-3 and Task-4.

## 5 CONCLUSION

In this paper, we introduce the DRAM-like architecture with an asynchronous refreshing strategy to effectively extract relations. The DRAM-like architecture is composed of a perceptron, a controller, and a refresher, and dynamic memory sample allocation is achieved through the cooperation of all three, which solves the conflict between relational memory imbalance and unbiased allocation of memory samples in previous work. We also propose a compromise asynchronous refresh strategy to find a pivot between over-memorization and overfitting, which concentrates on current tasks and asynchronously training mixed-memory data. Additionally, we provide a theory to explain existing refresh strategies and categorize them into centralized, distributed, and hybrid refreshing. The experimental results demonstrate the promise of our approach in CRE scenarios for effectively alleviating catastrophic forgetting. In future work, we will explore the effects of asynchronous refresh mechanisms and DRAM-like architectures in other memory-based continual learning tasks.

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

# A  EXTERNAL RESULTS

## A.1  Ablation Study

The extended experiments of EDRA and ConPL on the TACRED dataset are supplemented here, as shown in Tab. 5, which is consistent with the results in the main text.

**Table 5: Ablation experiments on the FewRel benchmark and the TACRED benchmark are used to validate the effectiveness of increasing the memory sample size and the asynchronous refreshing mechanism, The individual symbols represent the same meaning as in Tab. 1 and asyn denotes the asynchronous refreshing added to the method.**

| Method | Task Index | | | | | | | |
|---|---|---|---|---|---|---|---|---|
| | T1 | T2 | T3 | T4 | T5 | T6 | T7 | T8 |
| 10-way-5-shot of FewRel | | | | | | | | |
| ERDA[27] | 92.80 | 76.90 | 67.83 | 62.70 | 58.70 | 55.78 | 51.11 | 51.39 |
| ERDA$^\dagger$[27] | 92.20 | 76.50 | **71.00** | 61.30 | 63.30 | 57.00 | 53.99 | 54.13 |
| **EDRA$^\dagger$[27] + asyn** | **93.80** | **78.55** | 69.03 | **66.40** | **65.42** | **61.55** | **57.36** | **56.55** |
| ConPL[3] | 94.30 | 93.60 | 92.03 | 88.63 | 88.32 | 86.70 | 86.33 | 84.93 |
| ConPL$^\dagger$[3] | 95.90 | 93.70 | 92.20 | 88.70 | 88.34 | 87.33 | 87.60 | 86.74 |
| **ConPL$^\dagger$[3] + asyn** | **96.40** | **94.85** | **92.90** | **90.05** | **89.26** | **88.37** | **88.75** | **87.23** |
| 5-way-5-shot of TACRED | | | | | | | | |
| ERDA[27] | 70.08 | 41.30 | 39.40 | 37.58 | 29.18 | 23.08 | 20.97 | 21.21 |
| ERDA$^\dagger$[27] | 71.43 | 45.73 | 42.53 | **43.37** | 33.66 | 28.40 | 22.12 | 22.01 |
| **ERDA$^\dagger$[27] + asyn** | **75.10** | **49.53** | **44.38** | 39.33 | **36.04** | **29.10** | **24.12** | **23.66** |
| ConPL[3] | 97.37 | 84.68 | 81.54 | 79.77 | 81.21 | 76.96 | 74.49 | 73.23 |
| ConPL$^\dagger$[3] | **98.25** | 84.46 | 80.92 | 80.68 | 80.93 | 76.17 | 76.33 | **75.12** |
| **ConPL$^\dagger$[3] + asyn** | 97.37 | **85.78** | **81.64** | **81.59** | **81.98** | **79.27** | **76.60** | 74.93 |

