# OpenReview forum: "DRAM-like Architecture with Asynchronous Refreshing for Continual Relation Extraction"
_ACM.org/TheWebConf/2024/Conference — TheWebConf24 Oral_

### Official Review · Reviewer_QqUN · 2023-11-16

**Novelty:** 4
**Technical Quality:** 4

**Review:**

This paper proposes a new architecture for continual relation extraction (CRE) that uses an asynchronous refreshing strategy to tackle catastrophic forgetting. It is inspired by Dynamic Random Access Memory (DRAM) mechanisms and consists of two main components: a DRAM-like architecture containing three other subcomponents (perceptron, controller, and refresher), and the asynchronous refreshing strategies. Both components are carefully described in the paper, consolidating its two main contributions. It also evaluates the approach against two established benchmarks and three baselines. The numbers reported are not that impressive to me, as the current approach only improves the baselines by about 1.5%. I appreciate the ablation studies reported in Section 4.5, as they allow all component contributions to be effectively understood.

Although the paper is easy to read in general, it also shows some rare sentences and minor typos that must be checked. A revision of the language could help to improve the quality of the final manuscript. On the other hand, I recommend the author to rewrite the Conclusions section, because the current one is like a clone of the abstract, even it provides less information. I think they are missing the opportunity to show the impact of the proposal in a more assertive way, which would perhaps help a reader like me to see more clearly the contribution of the article, which in principle describes an interesting approach but with results that do not seem a substantial improvement on the state of the art.

**Questions:**

The authors do not provide any resources to facilitate the reproducibility of their experiments. Have they considered doing so in the future? What requirements should the platform meet in order to use the current approach?

**Ethics Review Description:**

I have not check "Yes" to the ethics review flag.

**Reviewer Confidence:**

1: The reviewer's evaluation is an educated guess

**Scope:**

3: The work is somewhat relevant to the Web and to the track, and is of narrow interest to a sub-community

---

### Official Review · Reviewer_nScK · 2023-11-21

**Novelty:** 6
**Technical Quality:** 6

**Review:**

The paper proposed a new method called Dynamic Random Access Memory (DRAM) for Continual Relation Extraction. It is inspired from DRAM update as physical memory. Its feature is to allocate memory dynamically to consolidate well-remembered relations and also to allocate additional memory for poor learned relations. It also includes asynchronous refreshing strategy to find the pivot between over-memorization and overfitting.
Apart from analogy of DRAM, the proposed method is a well-considered and practical method to solve the problems for CRE. The authors tested it by comparing the existing methods and claimed that it is superior than them. Its performance is constantly better than others through the series tasks mostly. The experiments and their evaluation are fairly so seem reliable.

**Questions:**

In Table 4, the forgetting performance is shown. It is interesting, but it only shows up to T5. It is possible to show the result until T8?

**Reviewer Confidence:**

2: The reviewer is willing to defend the evaluation, but it is likely that the reviewer did not understand parts of the paper

**Scope:**

4: The work is relevant to the Web and to the track, and is of broad interest to the community

---

### Official Review · Reviewer_tyxL · 2023-11-22

**Novelty:** 6
**Technical Quality:** 6

**Review:**

# Summary

The paper presents an approach for continual relation extraction. The framework aims to overcome the challenge of catastrophic forgetting via a novel architecture inspired by the mechanisms of Dynamic Random Access Memory, called DAAR. The approach has been evaluated on two benchmarks (FewRel and TACRED) and compared with three baselines (EMAR, ERDA and ConPL). Performance is measured in terms of the overall accuracy.

# Significance

Automatic extraction of relations from text is of paramount importance for streamlining the knowledge graph construction problem. In an era where textual content is continuously produced, attention has to be paid to the continual extraction problem and avoiding problems such as catastrophic forgetting. Similar problems have been addressed in computer engineering and therefore, the architecture presented is a direction worthwhile of further investigations.

# Relevance

It is relevant as relation extraction is a technique for constructing knowledge graphs.

# Readability

I’m not familiar with learning architectures and I found Section 3 (the section presenting the approach) hard to read. The authors assume readers are very familiar with the topic and this is necessary the case, since the conference is on web technologies and the track attains to semantics and knowledge. Maybe the readability can be improved by giving intuitions, examples and additional explanations of concepts like the Prototype (what is that? Can you provide the readers with an example? Why do we need it?), Perceptual score, memory etc.. After reading a couple of times the section I have only a vague intuition of the meaning of these concepts. In other words, the mathematics is clear, but the design rationale is vague. An important question is why do we need this component? The answer has to be clear to the reader when the component is introduced. I couldn’t find it in the text.

# *Novelty*

The problem is not original, but the approach is novel.

# Positioning wrt state of the art.*

To my knowledge, most of the relevant work is cited and positioned to the authors' contribution.

# Potential impact.*

Again, as I’m not very familiar with the research on this topic, so, I’m not sure how to assess the significance of the improvement of 1.5% in accuracy over the baselines. I’d value the impact as limited, but maybe this constitutes a big leap in the field.

# Technical soundness.

The approach appears sound and reasonable to me.

# Reproducibility.

Although the architecture is described in detail, the lack of source code hinders the reproducibility of the experiments.

**Questions:**

See the review

**Ethics Review Description:**

-

**Reviewer Confidence:**

1: The reviewer's evaluation is an educated guess

**Scope:**

3: The work is somewhat relevant to the Web and to the track, and is of narrow interest to a sub-community

---

### Official Review · Reviewer_EPcd · 2023-11-22

**Novelty:** 5
**Technical Quality:** 6

**Review:**

This paper looks at the problem of relation extraction in a continual learning step where new sets of relations are introduced over time. The aim then is to be able to extract new relations learnt at particular time without forgetting about relations learned earlier. This is a fairly widely studied task in NLP. This paper looks at a new way of implementing external memory for such continual learning systems that focuses on updating the memory from previously seen examples. The paper makes an analogy to physical computing memory (i.e. DRAM chips) to focus on I what I would term a variant of rehearsal learning - which is common in continual learning setups. The variant is to refresh the memory in dynamic fashion depending on a separate

The results show better performance against other methods in this domain on two relation extraction databases converted to be used in a continual learning setup.

Strengths
- Slightly better performance than other SOTA methods
- Interesting ablation studies
- Asynchronous refreshing  of memory looks to be unique in continual learning

Weaknesses
- The paper makes a lot of its analogy to physical DRAM chips but I think that's somewhat distracting to the reader when maybe more time should be spent just focusing on the importance of the asynchronous cycle
- It would have been interesting to see the potential bounds of relation extraction performance by just training a whole model not in a continual setup.
- The connection to rehearsal learning could have been made clearer.
- The memory employed is fairly small.

__After Rebuttal__
The rebuttal addressed my main questions in particular the additional experiments I think help a lot and the clearer connection to rehearsal learning.

**Questions:**

How is your approach related to rehearsal learning? Is it a variant of that approach? If not how is it different?

**Ethics Review Description:**

no ethics concerns

**Ethics Review Flag:**

Yes

**Reviewer Confidence:**

3: The reviewer is confident but not certain that the evaluation is correct

**Scope:**

3: The work is somewhat relevant to the Web and to the track, and is of narrow interest to a sub-community

---

### Official Review · Reviewer_ppqz · 2023-12-15

**Novelty:** 7
**Technical Quality:** 6

**Review:**

The paper proposes an approach for continuous relation extraction with a particular focus on addressing the challenge referred to as "catastrophic forgetting". In particular, the goal is to dynamically allocate memory in such as manner that consolidates
of well-remembered relations while allocating additional memory for revisiting poorly learned relations.
The approach proposed is based on an architecture that is inspired by Dynamic Random Access Memory and comprises of a  perceptron, a controller and a refresher. In addition, an asynchronous refresh strategy is proposed to find a pivot between
over-memorization and overfitting, which concentrates on current tasks and asynchronously training mixed-memory data.
A comparative evaluation against a state-of-the-art approach using two benchmarks shows some promising benefits of proposed approach


+ relevant problem with interesting solution strategy.
+ intuition and rationale for approach are well motivated.
- some basic editorial improvements are needed throughout the paper.

**Questions:**

None

**Reviewer Confidence:**

2: The reviewer is willing to defend the evaluation, but it is likely that the reviewer did not understand parts of the paper

**Scope:**

4: The work is relevant to the Web and to the track, and is of broad interest to the community

---

### Decision · Program_Chairs · 2024-01-22

**Decision:**

Accept (Oral)

**Comment:**

This article considers the problem of relation extraction in continual learning steps, where a new technique is introduced for implementing external memory that can be dynamically updated. Results show slight improvements compared to existing approaches.

 All reviewers agree that this work is relevant to the Web Conference and solves a relevant problem with a novel solution.
 After discussion, it was agreed that this work deserves to be accepted.
 We do recommend the reviewers to include the suggested changes from the reviewers, such as the suggested editorial fixes and improvements to readability.